# Passive Immunization in Alpha-Synuclein Preclinical Animal Models

**DOI:** 10.3390/biom12020168

**Published:** 2022-01-20

**Authors:** Jonas Folke, Nelson Ferreira, Tomasz Brudek, Per Borghammer, Nathalie Van Den Berge

**Affiliations:** 1Department of Geriatric Medicine, University Hospital Essen, 45147 Essen, Germany; 2Research Laboratory for Stereology and Neuroscience, Department of Neurology, Bispebjerg-Frederiksberg Hospital, University Hospital of Copenhagen, 2400 Copenhagen, Denmark; tomasz.brudek@regionh.dk; 3Copenhagen Center for Translational Research, Bispebjerg-Frederiksberg Hospital, University Hospital of Copenhagen, 2400 Copenhagen, Denmark; 4DANDRITE-Danish Research Institute of Translational Neuroscience, Department of Biomedicine, Aarhus University, 8000 Aarhus, Denmark; nelson@biomed.au.dk; 5Department of Clinical Medicine, Aarhus University, 8000 Aarhus, Denmark; borghammer@clin.au.dk; 6Department of Nuclear Medicine and PET, Aarhus University Hospital, 8200 Aarhus, Denmark

**Keywords:** alpha-synuclein, passive immunization, disease stratification

## Abstract

Alpha-synucleinopathies include Parkinson’s disease, dementia with Lewy bodies, pure autonomic failure and multiple system atrophy. These are all progressive neurodegenerative diseases that are characterized by pathological misfolding and accumulation of the protein alpha-synuclein (αsyn) in neurons, axons or glial cells in the brain, but also in other organs. The abnormal accumulation and propagation of pathogenic αsyn across the autonomic connectome is associated with progressive loss of neurons in the brain and peripheral organs, resulting in motor and non-motor symptoms. To date, no cure is available for synucleinopathies, and therapy is limited to symptomatic treatment of motor and non-motor symptoms upon diagnosis. Recent advances using passive immunization that target different αsyn structures show great potential to block disease progression in rodent studies of synucleinopathies. However, passive immunotherapy in clinical trials has been proven safe but less effective than in preclinical conditions. Here we review current achievements of passive immunotherapy in animal models of synucleinopathies. Furthermore, we propose new research strategies to increase translational outcome in patient studies, (1) by using antibodies against immature conformations of pathogenic αsyn (monomers, post-translationally modified monomers, oligomers and protofibrils) and (2) by focusing treatment on body-first synucleinopathies where damage in the brain is still limited and effective immunization could potentially stop disease progression by blocking the spread of pathogenic αsyn from peripheral organs to the brain.

## 1. Introduction

Twenty-five years ago, it was found that aggregated alpha-synuclein (αsyn) is the major protein component of Lewy pathology [1]. Subsequent studies discovered that point mutations within or duplications/triplications of the αsyn gene (*SNCA*) are linked to familial PD [2,3,4]. These findings indicate a central role of αsyn in Lewy body diseases (LBD). Since then, Parkinson’s disease (PD), dementia with Lewy bodies (DLB), pure autonomic failure (PAF) and multiple system atrophy (MSA) are classified as synucleinopathies, also called α-synucleinopathies, as they all are characterized by pathological accumulation of the protein αsyn. PD, DLB and PAF predominantly present with intraneuronal and neuritic deposits of misfolded αsyn, i.e., Lewy bodies and Lewy neurites. Furthermore, the accumulation of pathogenic αsyn is associated with progressive disrupted cellular function, neuronal death and subsequent dysfunction in the central and peripheral nervous system [5]. MSA is a distinct case of α-synucleinopathies, as it is characterized by predominant glial cytoplasmic inclusions (GCIs) [6], later also called Papp-Lantos bodies [7].

Patients are classified as PD, DLB, PAF or MSA based on their clinical symptoms and later, post-mortem by the spatiotemporal distribution of pathogenic αsyn [8]. The spatiotemporal distribution is likely dependent on a combination of different factors, disease onset site and neuroanatomical connections as well as cellular vulnerability and the presence of concomitant tau and/or Aβ pathology. The clinical representation of PD, DLB, PAF and MSA patients is highly heterogeneous esp. in early disease stages, and displays a large clinical overlap, as each α-synucleinopathy may include a wide range of motor, cognitive, gastrointestinal and/or other autonomic disturbances, complicating early and accurate diagnosis. For example, DLB merely differentiates from PD diagnosis by the occurrence of cognitive dysfunction prior to motor dysfunction by only one year [9], which is very short, considering that non-motor symptoms occur up to 20 years prior to motor symptoms in PD [10]. PD, DLB and MSA show both central and peripheral nervous system involvement of αsyn pathology [11,12]. In PAF, αsyn pathology is confined within the autonomic nervous system (ANS) without motor dysfunction [13]. These patients also have an increased risk to pheno-convert into other α-synucleinopathies later in life, possibly indicating a pathophysiological disease continuum [12]. Furthermore, MSA patients with autonomic-only presentation in the early disease stage can be misdiagnosed as PAF. Moreover, MSA patients presenting with parkinsonism may be misdiagnosed as PD [14]. These α-synucleinopathies progress at different velocities with different intensities, but may evolve to similar advanced disease stages over time where the entire body is affected [15,16].

Currently, there is no cure for any of these α-synucleinopathies; hence, there is a great interest in targeting pathogenic αsyn as a strategy to halt disease progression. To reduce levels of harmful misfolded αsyn, a clearing process of the protein has to be established. This can be achieved with immunotherapies using vaccination strategies with antibodies directed against harmful αsyn [17]. The aim of a particular immunotherapy is to reduce the amount of misfolded αsyn in the body, and thereby block the spread of pathogenic αsyn, consequently reducing progressive neurodegeneration and, therefore, symptoms [18]. Passive immunization with naturally occurring autoantibodies (nAbs) that are part of the innate immune system is considered more safe than active immunization or vaccination where an antigen is injected to induce the production of antibodies [19]. Preclinical studies using nAbs have shown reduced trans-synaptic spread of pathogenic αsyn, as well as improved motor and cognitive deficits in PD mouse models. In contrast, preliminary data from on-going clinical phase I and phase II trials using passive immunotherapies targeting different forms of αsyn are unable to demonstrate efficacy in reducing disease progression [20]. Whether nAbs provide protection against developing PD, increasing evidence suggests that anti-αsyn nAbs may have a protecting effect in inhibiting αsyn seeding and can recognize Lewy body pathology [21]. nAbs have been extensively evaluated in PD as reviewed by Scott et al. [22]; however, most studies have been restricted to assessing total IgG nAbs levels. A few studies have evaluated IgG nAb subclasses, IgM nAbs and the binding properties of these nAbs, showing a switched immunological response in PD and MSA patients and further a reduced binding towards αsyn [23,24,25]. A more thorough evaluation is needed to fully map the immunological responses in PD and other synucleinopathies.

Discrepancy between animal and patient studies might be explained by a combination of poor αsyn targeting and poor patient selection. The strain hypothesis in α-synucleinopathies postulates that each disease entity is characterized by a distinct conformation of pathogenic αsyn; therefore, each α-synucleinopathy could be caused by a unique αsyn structure or strain. This implies that different α-synucleinopathies require different nAbs targeting a specific αsyn strain. Unfortunately, clinical trials lack accurate patient stratification and individual disease heterogeneity is often not considered during patient recruitment, as trials assume a common pathogenetic mechanism of disease across patients. The highly heterogeneous profile of the prodromal disease phase of α-synucleinopathies make early and accurate stratification very challenging. Consequently, patients are often misdiagnosed at early disease stages and may not benefit from a certain immunotherapy. Further, patients in advanced disease stages with established major neurodegeneration might benefit less compared to prodromal patients. It remains to be elucidated whether the formation of mature dense αsyn or Lewy pathology aggravates or protects against neurodegeneration [26]. It is hypothesized that endogenous αsyn goes through four stages to ultimately form mature Lewy pathology: misfolding of endogenous αsyn, oligomerization, formation of fibrils and, finally, development of dense inclusions. The immature oligomeric and fibrillary αsyn appear to be most toxic compared to mature Lewy pathology [27], indicating such conformers could be particularly attractive as therapeutic targets instead of mature Lewy pathology. Lack of these considerations might have contributed to disappointing results. Future trials should focus on enrolment of prodromal patients after detailed stratification into different disease subtypes by using disease- and strain-specific biomarkers. Additionally, target biology should be optimized towards immature strain-specific pathology. For this purpose, it is crucial to gain insight in the earliest physiological to pathological events underlying αsyn misfolding and abnormal aggregation using animal models of α-synucleinopathies. Here, we discuss recent developments of passive immunization in animal models of α-synucleinopathies, their shortcomings and highlight the potential utility of novel experimental models and considerations for future clinical trials to increase translation ability of results.

## 2. αsyn and Its Role in the Pathogenesis of Synucleinopathies

αsyn was first discovered in 1988 in the Common torpedo (an electric ray species). It is an abundantly expressed protein in the brain, located in the presynaptic nerve terminals [28]. In humans, αsyn is a 140 amino-acid protein encoded by the *SNCA* gene and is part of the synuclein protein family including beta(β)- and gamma(γ)-synuclein [29]. αsyn has three domains: a N-terminal domain (residue 1–60); a central hydrophobic domain (residue 61–95), also called the non-amyloid-β component (NAC), and a negatively charged C-terminal domain (residue 96–140) [30]. αsyn exists in equilibrium between soluble cytosolic and membrane-associated forms. The N-terminal domain adopts an α-helical structure facilitating lipid membrane interaction [30]. This membrane association of αsyn most probably contributes to synaptic trafficking, accelerating vesicle reuptake by promoting membrane curvature [31,32]. However, several gaps of knowledge in the precise role of αsyn need to be elucidated. The native form of αsyn is a monomer, but due to its soluble state as an intrinsically disordered protein, αsyn is prone to self-assemble or misfold into a variety of insoluble oligomeric species. Studies have shown that the conditions are pivotal for the oligomeric species, some more toxic than others [33], thus demonstrating the difference in LB or GCI formation.

αsyn is likely to exist both as unstructured monomers and helical oligomers [34,35]. However, the existence of free helical structures has been debated since recapitulation of helical oligomers could only be provided with addition of lipids [36], N-terminal acetylation [37] and N-terminal extension in lipid-free environment [35], which impact a more uniform aggregation of αsyn [38,39]. The central region, the NAC domain, corresponding to aa 71–82 is essential for misfolding and aggregation [40]. What precise mechanisms initiate aggregation and determine the end-stage polymorphic structure within multimers and fibrils, still needs to be resolved. Since the discovery of mutations in a familial phenotype of PD, several other genetic alterations have been directly linked to PD. All familial PD related mutations in the *SNCA* gene translated into missense mutations in the protein (A30P, E46K, H50Q, G51D, and A53T), are all located in the N-terminal and clustered around the αsyn protein loop [41], and further found to alter oligomerization [42]. It is now believed that the aggregation of αsyn is the centralizing micro pathological change in α-synucleinopathies. How αsyn drives PD pathology remains elusive, and it remains to be elucidated whether the formation of mature Lewy bodies is neuroprotective or a facilitator of neurodegeneration [26]. However, recent findings implicate that the presence of oligomeric αsyn is the primary cause of neurotoxicity and plays a critical role in PD pathophysiology and propagation of pathology [43]. Importantly, the development of PD pathology and associated neurodegeneration is most likely a combination of several risk factors that, besides the formation of pathogenic αsyn, also include neuroinflammation, failing proteostasis mechanisms, mitochondrial dysfunction, endoplasmic reticulum stress, and synaptic and cell impairments.

Based on Braak’s theory [44], later updated by others [45], the pattern of αsyn spread in PD is divided into six successive stages; starting from early Lewy neurite (LN) lesions in non-dopaminergic structures of the lower brainstem, e.g., the dorsal motor nucleus of the vagus nerve in the medulla, ascending to the brain or from the olfactory nuclei to the cortex prior to symptomatic manifestations in stage 1, to more widespread LBs in the basal ganglia structures, manifesting early motoric symptoms with pale precursors of LBs in the substantia nigra pars compacta (SNpc) in stage 2. From Braak stage 3 to 4, αsyn lesions (LNs and LBs) propagate with further neuronal depletions in the amygdala and nucleus basalis of Meynert to severely damaged structures in the SNpc. In the late stage of PD (stage 5), the LB’s from the mesocortex transmit to the temporal and prefrontal neocortices, while in the end stage (stage 6), lesions are affecting the primary sensory and motor areas, constituting the entire neocortex [44,45,46]. Later, the LB staging has been revised, with αsyn pathology starting in either the olfactory bulb or the enteric cell plexi [47]. Even more recently, studies have shown that αsyn pathology in preclinical PD may occur simultaneously in multiple regions of both the peripheral and central nervous system [48]. Interestingly, αsyn pathology has been observed in incidental LB disease, where individuals have no clinical Parkinsonian manifestations [48]. DLB patients share similar pathological changes to PD patients; however, since many PD patients develop dementia (PDD) as the disease progresses, the separation between PDD and DLB patients is currently being debated, also neuropathologically. Similar to PD patients, the majority of DLB patients show degeneration of dopaminergic neurons in the SNpc [49]. The main difference lies in pathological changes affecting the neocortex and limbic system in DLB compared to PD patients. Microscopically, DLB is presented with widespread accumulation of LNs and LBs similar to the Braak staging system in PD. The consensus, however, divides DLB patients neuropathologically based on LB-related pathology, whether the distribution is centralized in the brainstem, limbic or neocortical regions [50]. Besides αsyn pathology, coexistent Alzheimer’s disease (AD) pathology, sufficient for a secondary diagnosis of AD, is observed in about 50% of DLB patients. The pattern of coexistent AD pathology is usually not associated with the αsyn propagation pattern [51].

In MSA, the main lesion pathology is found as GCIs [6], later confirmed to be made up primarily of αsyn [52]. Compared to incidental LB pathology, GCIs are rarely, if ever, observed in aging individuals without clinical symptoms. Thus, the definite diagnosis of MSA relies primarily on the presence of widespread GCIs [53]. Besides GCIs, MSA patients are also presented with glial nuclear inclusions, neuronal cytoplasmic inclusions and neuronal nuclear inclusions [54]. Since MSA patients are characterized by low-density spread of LBs, MSA is not considered a LB disease [55]. Studies showed that αsyn pathology in GCIs differ in post-translational permutations resulting in cell type specific insoluble conformational structures [56]. MSA patients are divided into two subgroups depending on their clinical manifestations and macro/microscopic pathology: MSA-P with predominantly parkinsonian symptoms and MSA-C with cerebellar symptomatology [53]. In fact, the αsyn burden of pathology projections seems to be highly associated with the subtype of MSA [57]. In addition to MSA-P and MSA-C there are several other subtypes of MSA, including minimal change MSA, non-motor MSA and incidental MSA (i.e., GCI pathology in the absence of clinical features) [58]. In the case of minimal change MSA, there is widespread GCI pathology but without clear corresponding neuro- or oligodendrogliopathy and a general absence of other clinical signs. Minimal change MSA can be interpreted as a coincidental, but distinct subtype, with diffuse LBD [59].

PAF is a very rare sporadic neurodegenerative disorder characterized by failure of the autonomic system. LBs in the peripheral autonomic nervous system underlie the majority of pathology in PAF. αsyn accumulates in the form of LBs within the sympathetic ganglia and in axons of autonomic neurons e.g., heart, bladder, skin and colon [60]. Occasional LBs in the SNpc and locus coeruleus have been reported [60], indicating pheno-conversion to PD. The precise and low-density widespread αsyn accumulation in PAF remains unclear.

## 3. Prion-like Behavior and Gut-to-Brain Propagation

αsyn is unequivocally linked to neurological disease. However, the pathophysiological mechanisms underlying α-synucleinopathies are still unknown. A seminal study in two PD patients showed that surviving transplanted fetal nigral neurons developed Lewy bodies over time, indicating pathogenic αsyn from a patient’s brain is able to spread to healthy grafted neurons [61,62]. The ability of pathogenic αsyn to convert normal endogenous αsyn protein into pathogenic misfolded αsyn (also called ‘seeding’), and subsequently propagate to a neighboring neuron is called conformational templating. The phenomenon of conformational templating was first discovered in prion diseases where a pathogenic seed recruits cellular prion protein (PrPc) and converts it into a toxic isoform called a prion (PrPSc) [63,64]. Similar to prions, oligomeric αsyn species are also able to template or seed intracellular aggregation of αsyn in vitro [33]. In α-synucleinopathies, it has been shown that misfolded αsyn can convert into toxic isoforms such as oligomers that accumulate into fibrils. These fibrils are capable of crossing the neuronal membrane and once transferred into a new cell, can convert normal αsyn into misfolded αsyn that further evolves into oligomeric and fibrillar αsyn, hereby initiating an auto-replicating process [26].

Besides in the brain, αsyn pathology is also observed in several peripheral organs of PD and MSA patients [9,10] including the gut [65,66,67], heart [68] and skin [69,70,71]. Some of these studies observed pathology in the gut [65,72] or skin [12,69,73] at early prodromal disease stages, demonstrating its potential as early disease biomarker. Additionally, non-motor autonomic symptoms, including constipation, orthostatic hypotension, pain, urinary and sweating problems are common in early PD and MSA, and occur up to 20 years prior to motor symptoms [10,74,75,76]. These findings support that pathogenic αsyn can transmit from cell to cell within the central nervous system (CNS), as well as from the peripheral nervous system (PNS) to the CNS in a prion-like fashion. Several preclinical studies have shown mechanistically that pathogenic αsyn is able to template and spread trans-synaptically in a prion-like fashion (reviewed in [77,78]). Interestingly, it has been shown that αsyn propagation within sensory afferents is concomitant with impaired nociceptive response, reflected by mechanical allodynia, reduced nerve conduction velocities (sensory and motor) and degeneration of small- and medium-sized myelinated fibers, suggesting that αsyn trans-neuronal transmission and conformational templating might underlie the multifaceted etiology and symptomatology of pain in PD [79].

Despite extensive investigation, mechanisms inducing αsyn misfolding are still unclear. It has been suggested that pathogenic αsyn can initiate in the gut due to exposure to environmental toxins that enter the gut lumen, affecting gut permeability and oxidative stress via endocrine cells in the gut epithelium, ultimately leading to the formation of misfolded αsyn in the myenteric plexus, and spread to the brain via peripheral nerves [80]. Importantly, oral gavage with the pesticide rotenone in old mice induced formation of misfolded αsyn in parasympathetic and sympathetic nuclei of the CNS [81]. The gut-to-brain transmission of pathogenic αsyn is difficult to study in living patients. However, recent experimental animal models of PD have been able to recapitulate this aspect of PD pathogenesis. Researchers have shown that injections of pathogenic αsyn in the gut of rodents induces pathogenic αsyn in brainstem structures from 1 month post injection, and subsequently induces a caudo-rostral spread of pathology [82,83], paralleled by loss of dopaminergic neurons as well as motor and or non-motor symptoms resembling idiopathic PD [83,84]. Interestingly, pathogenic αsyn did not ascend into the brain when animals underwent truncal vagotomy [84]. Also in humans, it has been shown that truncal vagotomy reduces the risk of PD by 40–50% after 10–20 years of follow-up, indicating that cutting the vagal nerve inhibits the spread of αsyn pathology to the brain [85,86]. Additionally, bidirectional gut-to-brain and brain-to-gut propagation of pathology along the vagus nerve was observed post injection of pathogenic αsyn in the gut of rodents, as well as propagation to the heart and skin along sympathetic nerves, like observed in human PD [82,83]. Therefore, these models probably provide the most complete recapitulation of human PD to date. Taken together, these findings suggest that strategies aimed at prevention of cell-to-cell and gut-to-brain transmission of αsyn could slow down or halt transmission of toxic αsyn confomers and progression of symptoms in synucleinopathies, and that these models would be most suitable for developing such treatment strategies.

## 4. Subtypes of α-Synucleinopathies

The hypothesized gut-to-brain spread of αsyn in PD (and other synucleinopathies) is still heavily debated, since several autopsy studies could not confirm the proposed caudo-rostral spread and showed that αsyn pathology in the CNS is quite often present without the occurrence of αsyn in the ENS, vagus nerve or dorsal motor nucleus of the vagus nerve [87,88]. Overall, post-mortem studies have reported two principal types of pathology patterns: a brainstem-predominant type with more pathology in the brainstem than more rostral structures and a limbic/amygdala-predominant type, with more midbrain pathology compared to more caudal structures [16,89]. This observation has led to the body-first vs. brain-first hypothesis, a hypothesized stratification of PD subtypes based on the onset site of pathology, first described in 2019 by Borghammer and Van Den Berge [90]. The body-first type is associated with a brainstem-predominant pathology pattern in the brain, and is characterized by REM sleep behavior disorder (RBD) in the pre-motor phase and autonomic dysfunction (orthostatic hypotension and pathological 123I-metaiodobenzylguanidine (MIBG) heart scintigraphy) prior to motor dysfunction [91], as well as gut and skin tissue biopsies positive for aggregated αsyn [72,73]. Isolated RBD is a common finding among patients with synucleinopathies, usually coincides with other autonomic disturbances and is considered to be an early stage of (body-first) α-synucleinopathy [92,93,94]. In contrast, the brain-first type is hypothesized to be associated with a limbic/amygdala-predominant pathology pattern in the brain, and is characterized by an RBD-negative prodromal phase and nigrostriatal dopamine deficit prior to autonomic dysfunction. Both disease subtypes converge to a homogeneous advanced disease stage over time where the entire brain and several peripheral organs are affected. The existence of these subtypes is supported by in vivo imaging studies of isolated RBD and *de novo* PD patient groups. Isolated RBD and *de novo* PD with RBD (i.e., body-first subtype) were characterized by cardiac and enteric denervation, measured with MIBG SPECT and donepezil PET, respectively, and a relatively normal brain scan. In contrast, PD patients without RBD (i.e., brain-first subtype) were characterized by nigrostriatal neurodegeneration measured with FDOPA PET and a less pathological heart and gut scan, indicating damage to the brain precedes autonomic damage in the brain-first subtype, and vice versa in the body-first subtype [91].

In extension to the body-first vs. brain-first hypothesis, it has been recently postulated that the clinical representation and distribution of αsyn pathology in those two PD subtypes could be explained by varying disease onset site, body or brain, as well as the neural connectome [15]. According to this αsyn Origin site and Connectome (SOC) model, in the brain-first subtype, αsyn pathology arises in a single hemisphere of the brain, leading to asymmetric limbic-predominant pathology in the brain, after which it spreads to the body. According to the SOC model, in the body-first subtype, αsyn pathology arises in any peripheral organ (usually the gut), after which it spreads bilaterally, via overlapping vagal innervation, to the brainstem, leading to a bilateral and more symmetric brainstem-predominant pathology distribution in the brain [15]. Recent imaging studies [91] and neuropathological evidence [16] are in support of this SOC model. Nigrostriatal degeneration measured with FDOPA PET and DaT SPECT was significantly more symmetric in patients with iRBD and *de novo* PD with RBD versus PD patients without RBD. These data support that body-first PD is characterized by more symmetric distribution, most likely due to more symmetric propagation of pathogenic αsyn within the brain, compared to brain-first PD [95]. The SOC model actually applies to all LBD, including DLB and PAF patients [15].

In addition, validity of the SOC model is observed in animal models of PD, with symmetric involvement of the lower brainstem initially, upon peripheral initiation of pathology (gastrointestinal, intravenous, intraperitoneal or oral). This is followed by progressive symmetric involvement of the substantia nigra, limbic, and cortical regions in a predictable fashion. In contrast, unilateral initiation of pathology in the brain causes progressive predominant ipsilateral CNS involvement that is 3 to 10 fold higher than in the contralateral hemisphere. The pattern of CNS pathology (and symptoms) upon intracerebral initiation is highly dependent on the location of the first pathology. Therefore, brain-first PD is more heterogeneous compared to body-first PD [96].

## 5. Therapeutic Strategies Targeting αsyn Pathology

Several mechanisms either neutralizing or facilitating clearance of toxic pathological αsyn species have been suggested and attempted, and although there have been considerable advances, none to this date have been deemed successful in clinical trials [97]. Several approaches for modulating these toxic species have been proposed such as to reduce αsyn expression by lowering RNA levels [98,99], by increasing proteasomal or autophagy activity [100], by inhibiting oligomerization/fibrillation with small molecules [101] or by interference and clearance using immunotherapy. Immunotherapy is an appealing way, harvesting the immune system’s own resources in clearing potential toxic protein structures. There are two paths in pursuing immunotherapy: active immunization or passive immunization. Active immunization involves activation of the immune system to specific parts or conformational structures of αsyn. Most prominent in clinical trials with PD and MSA patients are active immunization using small 8-amino acid peptides, that mimic small regions of human αsyn called affitopes (PD01A and PD03A), developed by the company AFFiRiS [102]. Another clinical active vaccination trial is under way from United biomedical Inc., also using a C-terminal region-specific epitope [103]. Although less expensive than passive immunization strategies, one main obstacle is that it requires the patient’s immune system to actively recognize and activate the immunological cascade which triggers the formation of new antigen-specific B and T cells, and in consequence the antigen-specific antibodies. An important issue when exploring active vaccination strategies is the already existing autoimmunity. Evidence implicates that autoimmunity is a wide-ranging process of PD progression and recent studies suggest that T cells in certain PD haplotypes are auto-reactive towards αsyn [104,105]. Whether they recognize native or newly formed neoepitopes remains to be elucidated, and whether this autoimmunity serves as protective or pathogenic need to be investigated further. One important notion is that a potential active vaccination needs to overcome immune tolerance to self-antigens whilst avoiding autoimmunity.

As one of the main approaches of immunoregulatory therapies in α-synucleinopathies, passive immunization is based on continuous and chronic administration of αsyn- and/or conformation-specific antibodies, hereby recognizing toxic epitopes, facilitating clearance by innate immune cells, halting the spreading of pathogenic αsyn aggregates and potentially modifying disease progression. Although passive immunization cannot penetrate cell membranes, administration of αsyn antibodies could halt cell-to-cell transmission and clean up excess amounts extracellularly (see Figure 1). Moreover, passive immunization with continuous administration offers advantage in controlling several variable parameters during the treatment e.g., systemic antibody levels, bypassing uncontrolled T-cell stimulations and controlling adverse side-effects. Furthermore, specific antibodies can be engineered to function specifically and be raised against specific conformational epitopes. Even though passive immunization comes with several disadvantages such as high costs, time-consumption and side-effects such as hypersensitivity, it does not outweigh the advantages for patients. Although it has no overt disease symptoms, it could slow down the progression and retain the patient on hold in the prodromal phase with only few none-invalidating symptoms. Moreover, in more severe cases such as seen in MSA and DLB patients, it can severely improve life expectancy and quality of life.

## 6. Passive Immunization Strategies in Animal Models and Clinical Trials

Numerous passive immunization strategies have and are currently being tested in preclinical animal models (Table 1).

### 6.1. C-Terminal Targeting Approaches

The first candidate antibody tested in preclinical models was the monoclonal antibody (mAb) clone 9E4 targeting the C-terminal of human αsyn [106,120]. The 9E4 murine mAb recognizes the amino acids (aa) 118–126 of human αsyn (hαsyn) and has been shown to reduce toxic truncated species of αsyn, rescued behavioral deficits in PD-GFβ-αsyn transgenic mice and co-localizes with pathology in several brain regions [106]. These results were confirmed again by Masliah’s group [18] in the Thy1 αsyn (line 61) mice, further expanded to investigate the 9E4 analogs, the 5C1 and 5D12, and the 1H7 targeting the overlapping region of NAC and C-terminal. The 1H7 and 5C1 showed comparable decreased toxic αsyn truncated species, proposed to be reduced by internalization and lysosomal degradation [106], as well as improved behavioral deficits and protected tyrosine hydroxylase (TH) cell loss [18]. The 1H7 mAb was further investigated in Thy1 αsyn (line 61) mice, laterally injected with human αsyn expressing Lentivirus [112]. The 1H7 reduced axonal aggregation of αsyn and protected axonal integrity, as well as improved memory deficits and increased colocalization of αsyn and Iba-1 positive microglia, suggestive for microglia phagocytosis of extracellular αsyn [112]. The main difference is that 1H7 preferably binds aggregated αsyn at the C-terminus but also monomers. Following the results of Masliah and colleagues [112,120], targeting the C-terminal has become an optimistic immunization targeting strategy. Thus, several other antibodies have been produced targeting the C-terminus of αsyn. Parallel to the 9E4 mAb, another mAb targeting the C-terminal, the Ab274, was additionally investigated in collaboration between Masliah, Seung-Jae Lee and colleagues [107]. The Ab274, a IgG2a murine mAb, was investigated in PD-GFβ-αsyn mice (line M) showing reduced αsyn in cortical and limbic brain regions by microglial phagocytosis, additionally improving behavioral deficits [107]. Two other mAbs have been produced to target the C-terminal of αsyn, the Syn211 [108] and AB2 [110]. The Syn211 was tested in wild-type (wt) mice with intrastriatal injection of preformed αsyn fibrils (PFFs) and reduced insoluble αsyn and phosphorylated αsyn aggregates [108]. The AB2 mAb similarly reduced αsyn in brain homogenates in nigral αsyn-overexpressing wt rats [110].

### 6.2. N-Terminal and NAC Targeting Approaches

Interestingly, Tran and Shahaduzzaman tested an N-terminal-targeting antibody in parallel: Syn303 (aa 1–5) [108] and AB1 (aa 16–35) [110]. It seemed that the mAbs targeting the N-terminal surpassed the effects of the C-terminal targeting mAbs. In addition to overall reduced αsyn levels, the Syn303 reduced αsyn spread in the SNpc with 30% and in the ipsilateral and contralateral amygdala with 40%, and further improved motoric deficits [108]. However, in a later study, Syn303 was found inferior to their novel syn9048 mAb targeting the C-terminal and preferably binding aggregated αsyn structures [116]. The N-terminal-targeting AB1 additionally reduced DA and NeuN cell loss [110]. Very recently Chen and colleagues [118] (Chen et al., 2021) conducted a preclinical study using a NAC-targeting mAb (NAC32), which showed reduced αsyn pathology in the SN (25%), prevented TH+ neuron degradation and further reduced behavioral deficits [118]. Targeting monomeric (soluble non-toxic) αsyn proposes a different challenge, as reduction of functional αsyn potentially could harm normal physiological properties. After all, studies investigating αsyn knock-out or knock-down have shown aberrant dopamine synthesis and release, and even dopaminergic degeneration [121], and potential other physiological functions. It is therefore of utmost importance to ensure that mAbs targeting monomeric non-toxic αsyn do not negatively affect normal dopamine synthesis and/or its release. A way to circumvent this challenge is to target extracellular toxic αsyn conformers.

### 6.3. Conformational Targeting Approaches

Numerous antibodies have been developed targeting different αsyn conformational structures, from small oligomeric to larger fibrillary structures. Lindstrøm and colleagues [109] were the first to report on a mAb selective for conformational αsyn structures, this mAB47 is an IgG1 mAb which only reduces αsyn protofibrils in the spinal cord, but not in the brain, of Thy-1-H[A30P] mice [122]. Kallab and colleagues [113] later worked with a different clone of mAB47, called Rec47, in an MSA mouse model, the PLP-αsyn tg mouse model, which, in contrast to Lindstrøm and colleagues [109], showed reduced microglia signal and reduced activated microglial cells, correlated to reduced oligomeric αsyn. Furthermore, they observed reduced GCIs in the spinal cord, colocalization of phosphorylated αsyn pathology and correlation between Iba-1 positive microglia and oligomeric αsyn. They suggested an autophagy-directed elimination of αsyn [113]. Very recently, Nordström and colleagues thoroughly investigated the mAb47 (murine version of ABBV-0805), firstly establishing the binding region of the mAb to the C-terminal (121–127 aa) of αsyn, but more selective for aggregated αsyn species [119]. Nordström and colleagues extensively evaluated mAb47 in three different PD mice models with and without injection of preformed fibrils (to induce seeding) in both a prophylactic and therapeutic manner. They observed in wt mice, as well as in Thy-1-h[A30P] mice injected with 10 µg fibrils in the gastrocnemius muscle, a prolonged survival with the mAb47 treatment. In a Thy-1-h[A30P] mice injected with 1 µg fibrils, they further observed a reduced soluble and insoluble αsyn in the brain and reduced levels of phosphorylated αsyn in the CSF in both a prophylactic and therapeutic regime. Moreover, both soluble and insoluble levels were reduced in the brain in a dose-dependent administration of mAb47, more effective towards soluble αsyn. Lastly, they investigated the efficacy of mAb47 in an A53T+/− intracerebral fibril-seeding mice model with fibril injection into the anterior olfactory nucleus. After 16 weeks of weekly mAb47 intraperitoneal administration, spreading of phosphorylated αsyn was reduced in the CA1 hippocampal region [119]. El-Agnaf and colleagues studied three antibodies selective for oligomers and aggregates (Syn-01, Syn-02 and Syn-04) and two for mature aggregates (Syn-F1 and Syn-F2) [111]. Weekly injections over a 3-month period in mThy1 αsyn (line 61) mice showed that the Syn01, Syn-04 and Syn-F1 exhibit an overall similar effect by reducing αsyn in central brain regions (striatum, SN, and neocortex). Moreover, they reduced total αsyn, oligomeric αsyn and Syn-01, Syn02 and Syn-04 also reduced 5G4-aggregated αsyn. Only the Syn-01, Syn-04 and Syn-F1 rescued neuronal degradation and behavioral deficits. Syn-01 and Syn-04 further reduced astro- and microgliosis [111]. As for the 1H7, Schofield and colleagues from AstraZeneca among others developed a high-affinity monoclonal anti-αsyn antibody, MEDI1341, which binds the C-terminal monomeric form and aggregated αsyn [114]. Weekly administration of MEDI1341 in mThy1 αsyn mice with intra-hippocampal αsyn injections [112], reduced αsyn in hippocampal and neocortical areas [114]. As mentioned, Henderson and colleagues [116] tested the preferred binding of the novel Syn9048 mAb. Comparable to the previously tested mAb, Syn303 [108], Henderson et al. demonstrated reduced spread of αsyn pathology in the brain and attenuated dopamine reductions in the striatum of wt mice with PFF unilateral injection in the dorsal striatum [116]. Huang and colleagues used a different approach, isolating anti-αsyn nAbs from IViG using column chromatography, and administered them weekly at low (0.8 mg/kg) and at high (2.4 mg/kg) dosages in a A53T transgenic PD mouse model [115]. In both low and high dosages Huang and colleagues showed that nAbs reduced phosphorylated αsyn and soluble αsyn in the brainstem. Both dosages reduced astrocytes in the striatum and increased αsyn and microglia co-localization, as well as rescued motoric deficits. The rescuing effects were shown to be effective in a dose-dependent manner, with further reduced phosphorylated αsyn in cortical areas and reduced total human insoluble, soluble and oligomeric αsyn as in the brainstem. The effect of higher dosage further rescued behavioral deficits, in addition to the rescuing effect of pathological alterations e.g., reduced activated microglia and rescued TH+ positive neurons among others [115]. The BIIB054, also called cinpanemab, is a monoclonal mAb targeting the N-terminal (aa 1–10) with 800-fold greater affinity towards aggregated αsyn produced by Weihofen and colleagues in collaboration between Biogen Ltd. and Neurimmune AG Ltd. [117]. Weihofen and colleagues tested the BIIB054 in three different mouse models: (1) in female wt seeded contralateral with fibrils, they observed reduced truncated αsyn at 100 days and improved hangwire test at 60 days; (2) in male transgenic A53T mice (M83) seeded with fibrils in the striatum, they showing less severe paralysis at day 5, reduced paralysis at day 7 and weight loss at day 9; and (3) in male and female fibril-seeded BAC αsyn A53T mice [123], they reported rescuing effects of the contralateral DAT signal at 90 days post seeding [117].

Huang and Weihofen investigated αsyn-specific IViG nAbs and the BIIB054 mAb respectively [115,117], and both incorporate the idea that healthy individuals have antibodies resisting pathology. Huang and colleagues isolated anti-αsyn nAbs from IViG, containing immunoglobulins gathered from a large healthy population [115]. Weihofen and colleagues went a step further, investigating the paratopes from a repertoire of B cell receptors (BCRs) from healthy individuals and produced αsyn-specific nAbs from the repertoire [117]. In both studies, the nAbs showed significant rescuing effects in preclinical animal PD models. Table 1 shows an overview of PD animal studies investigating the different passive immunization strategies.

### 6.4. Passive Candidates Translated into Clinical Trials

Of the preclinical evaluated passive immunization candidates, a few have been translated into clinical trials (Table 2). *Prasinezumab* (PRX002) is a humanized IgG1 antibody from the murine version of 9E4 [18,106]. Although it did not meet its primary outcome (MDS-UPDRS), the antibody significantly showed decline on the UPDRS-III and patients with fast progressive and severe symptoms benefited more from the treatment and is currently running phase II, the PASADENA study. The second antibody tested in clinical trials is the mAB47or rec47 [109,113], now called ABBV-0805, however, the company AbbVie cancelled the phase Ib trial due to strategic reasons. MEDI1341 from AstraZeneca and Takeda Pharmaceuticals are currently running its phase Ib in early PD patients; the study will run into 2022. BIIB054, also called *Cinpanemab*, classified as a human-derived mAb made through reverse translational engineering, started a large phase II study, SPARK, but halted the development of *Cinpanemab* after it missed its primary and secondary endpoint. A fourth mAb, called LU AF82422, a humanized IgG1 monoclonal antibody, did not report any preclinical report, and no results from its phase I study are available yet. However, they recently released a phase II initiation press release.

## 7. Towards Personalized Immunotherapy

Several mechanisms have been implicated to trigger the initiation of pathogenic αsyn in the gut. Besides regulating the uptake of nutrients and water, the gut also provides an essential barrier against harmful or toxic substances from the external environment entering the body. About 400 m^2^ of gut internal membranes are exposed to environmental factors, compared to ~2 m^2^ of total skin surface area, meaning the gut is the main organ protecting against exposure to foreign pathogens [124]. It has been shown that bacterial and environmental toxins that enter the gut lumen can cause disruption of the intestinal epithelial barrier [125], alter the gut microbiome [126] and cause mucosal inflammation and oxidative stress [127,128]. A complex interplay of these factors are then able to trigger αsyn misfolding in the gut plexi, and an increased permeability of the intestinal barrier or ‘leaky gut’ will ultimately provide a route of transmission for the gut-formed αsyn seeds to the brain [78]. These findings indicate the gut as an important target for passive immunization therapy for two reasons. Early intervention in prodromal disease stages of gut-first cases may halt formation of pathogenic αsyn and subsequent gut-to-brain propagation. Second, only 0.1–0.2% of nAbs cross the blood–brain barrier. Therefore, it is conceivable that immunotherapy in prodromal patients with ‘leaky gut’ could be more effective. Increased gut permeability in prodromal patients with leaky gut might yield a better uptake of the administered nAbs near the source of pathogenic αsyn, resulting in a better treatment efficacy, as opposed to brain-first cases where the source is located in the brain (see Figure 2). Body-first PD patients are characterized by a more rapidly progressing phenotype, with faster motor and non-motor progression and more rapid cognitive decline, compared to brain-first PD patients [15,90]. This might explain why patients with fast progressive and severe symptoms benefited most from the treatment with Prasinezumab in the clinical trial. The validity of the SOC model requires further investigation, esp. in the prodromal phase. Detailed phenotyping of non-iRBD prodromal (i.e., brain-first subtype) patients is not yet available. Therefore, fundamental questions remain to be addressed: how these subtypes differ in their disease initiation mechanisms and progression patterns (esp. in the prodromal phase), and how such knowledge could be exploited for tailored subtype-specific immunotherapy. Future animal models should take into account varying disease onset sites to obtain causal and mechanistic understanding of the body-brain link in different disease subtypes, and to discover subtype-specific targets for immunotherapy. Recently developed, more sensitive, investigative tools such as PMCA (Protein-Misfolding Cyclic Amplification), RT-QuIC (Real-Time Quaking-Induced Conversion), PLA (Proximity Ligation Assay) and thiophene-based assays should be included while studying synucleinopathies to investigate the relation between disease onset site and subtype-specific strain characteristics. The identification of subtype-specific αsyn aggregates in easily accessible peripheral fluids or tissues from brain-first or body-first cases may enable early stratification as well as development of subtype-specific nAbs for immunotherapy.

Future clinical studies should focus on detailed imaging-based phenotyping for accurate stratification of prodromal disease subtypes, as careful patient selection for clinical trials will likely increase treatment efficacy and translation ability of preclinical studies. An αsyn PET tracer would allow for early stratification and detailed investigation and follow-up of synucleinopathy subtypes. Until that is discovered, a combination of other biomarkers should be used. The gut and skin, as well as blood and CSF, are easily accessible for biopsy studies to detect and quantify (subtype-specific) αsyn. Using ultra-sensitive methods, such as PMCA or RT-QuIC, on these biopsies, could contribute to an a priori screening of patients with toxic prion-like αsyn phenotype. This could provide not only more personalized interventions, but also plan for more effective clinical trials with minus-αsyn PD patients, as proposed to be the case in Parkin and LRRK2 mutation carriers [129]. Furthermore, in combination with imaging techniques, such as a DaT brain scan, MIBG heart scan and donepezil gut scan, this may enable prodromal diagnosis, together with quantification of non-motor symptoms such as RBD (polysomnography), gastrointestinal transit time (radio opaque markers [130], orthostatic hypotension and dementia (cognition test). Detailed imaging-based and αsyn templating-positive phenotyping is of significant importance to identify patients in the earliest phase of the disease, but also to evaluate treatment effects of immunotherapy (see Figure 2). Nevertheless, the road to use immune-based therapies on the basis of *a priori* preselected individuals is still long and cumbersome.

## 8. Future Perspectives and Conclusions

nAbs targeting soluble monomeric αsyn may affect dopamine synthesis negatively; therefore, it is preferred to target extracellular toxic insoluble αsyn conformers. Numerous antibodies have been developed targeting different αsyn conformers, ranging from small oligomeric to larger fibrillar structures. Preclinical studies have shown significant rescuing effects of nAbs treatment in PD animal models. Of the preclinical evaluated passive immunization candidates, a few have been translated into clinical trials, with suboptimal results, probably due to suboptimal patient selection with a mix of different PD phenotypes that are mainly situated in advanced disease stages. To improve treatment efficacy, a combination of imaging-based and histology-based biomarkers should be employed to identify body-first synucleinopathies in the earliest stages. Early immunization of these patients will prevent body-to-brain spread of pathology prior to irreversible dopamine damage in the brain. Current nAbs have been evaluated solely in brain-first animal models of synucleinopathies. In the future, animal models of body-first synucleinopathies should be employed to validate new and existing nAbs candidates. Such studies should investigate the potential of nAbs treatment to slow down or block accumulation of pathology in peripheral organs and subsequent spread through the entire autonomic connectome, as well as its potential rescuing effect on associated synaptic and neuronal dysfunction.

## Figures and Tables

**Figure 1 biomolecules-12-00168-f001:**
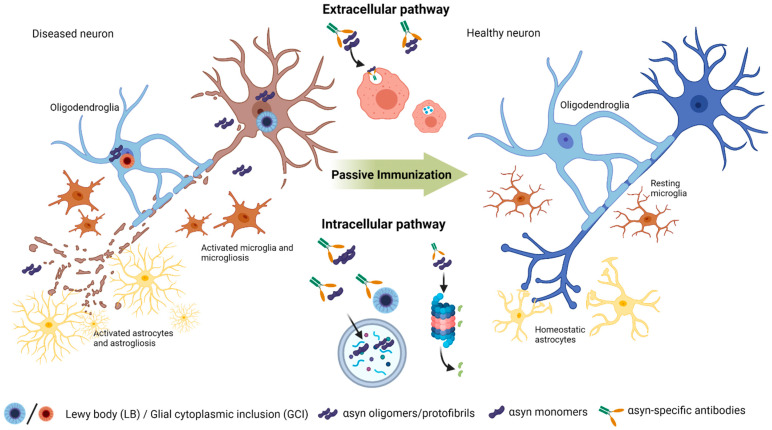
Clearing process of pathogenic alpha-synuclein (αsyn) using naturally occurring αsyn conformation-specific antibodies in passive immunization. In synucleinopathies, pathogenic αsyn species (proto/fibrillary or oligomeric) accumulate and potentially seed monomeric αsyn facilitating transmission, and additionally triggering microglial and astrocytic activation. In PD and DLB, αsyn aggregates in neurons form LBs. Whereas in MSA, αsyn accumulates in oligodendroglial cells, forming GCIs. Administration of αsyn-specific antibodies could facilitate clearance of pathogenic αsyn in the extracellular space by phagocytosis reducing transmission of pathogenic species or enable intracellular antibody-aided autophagy and proteosomic degradation, both pathways leading to reduced pathogenic αsyn and rescue of neuronal degradation. Created using Biorender.com (accessed on 30 November 2021).

**Figure 2 biomolecules-12-00168-f002:**
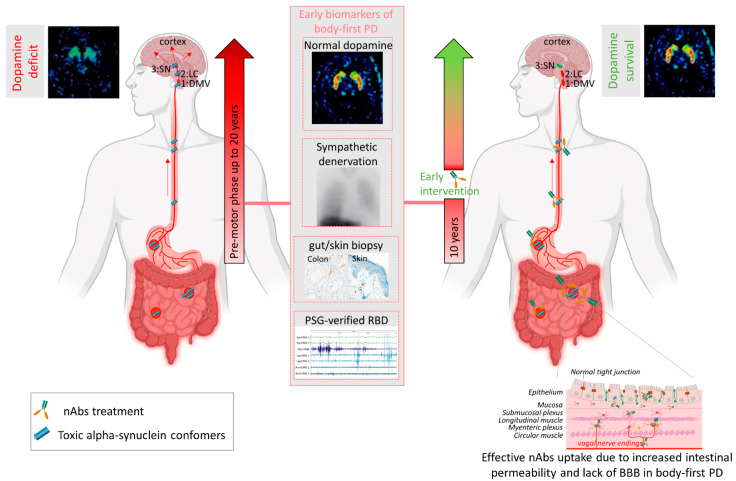
Passive immunization of pre-motor body-first PD patients enhances dopamine survival. Patients with probable prodromal body-first PD could be identified by a combination of several early biomarkers, such as the presence of pathological alpha-synuclein (αsyn) in skin and/or gut biopsies, polysomnography-verified RBD, cardiac sympathetic denervation on MIBG scintigraphies, but normal or near-normal nigrostriatal dopaminergic innervation on DaT SPECT. Such detailed phenotyping in the pre-motor phase might reveal body-first PD, allowing early intervention and optimal patient selection for clinical trials. Pre-motor start of nAbs treatment increases treatment efficacy by delaying or blocking peripheral-to-brain propagation of pathology, before any irreversible damage to the dopamine system is done, hereby enhancing the probability of dopamine survival in body-first PD. Furthermore, increased gut permeability in prodromal body-first PD patients with ‘leaky gut’ or increased intestinal permeability might yield a better uptake of the administered nAbs near the source of pathogenic αsyn conformers, resulting in a better treatment efficacy, as opposed to brain-first cases where the source is located in the brain and only 0.1–0.2% of nAbs cross the blood–brain barrier. Abbreviations: nAbs: naturally occurring autoantibodies; DMV: dorsal motor nucleus of the vagus; LC: locus coeruleus; SN: substantia nigra, PAF: pure autonomic failure, PSG: polysomnography, BBB: blood brain barrier. Created using Biorender.com (accessed on 30 November 2021).

**Table 1 biomolecules-12-00168-t001:** Passive immunization studies in Parkinsonian animal models.

Target (αsyn)	Antibody/Clone	Binding Site (aa)	Ab Origin Immunization Method	Injection Frequency Duration Amount	Animal Model	αsyn Pathological Effects	Neuronal Effects	Other Non-Neuronal Effects	Behavioral Effects	Ref.
C-term.	9E4 (IgG1)	C-term. 118–126	human Full-length (FL) αsyn (h-αsyn)	i.p., weekly 6 m 10 mg/kg b.w.	PD/DLB: PDGFb αsyn mice (line D)	↓ FL αsyn in neocortex neuropils; ↓ Reduced CC-αsyn in neocortex (intraneuronal and neuropil), Hippocampus (intraneuronal/neuropil); ↓ Reduced insoluble-FL-αsyn oligo.; ↓ soluble-CC-αsyn mono./oligo.; ↓ insoluble-CC-αsyn mono./oligo.	↑ synaptic densities; ↑ Pre-synaptic terminals; ↑ PSD95; ↑ Synapsin.	↓ Astrogliosis in PFC.	↑ Rotarod time; ↓ Path in Morris water maze.	[106]
C-term.	Ab274 (IgG2a)	C-term. 120–140	h-αsyn	i.p., weekly 4 wks 1 mg/mL hippocampal injection	PD: PDGFb αsyn mice (line M)	↓ αsyn (70–80% in cortex and hippocampus; ↓ αsyn (30–35% in striatum); ↓ αsyn in brain homogenates; ↓ αsyn in neurons and glial cells (total) (neocortex, Hippocampus, Striatum); ↑ Cathepsin-D and αsyn coloc.; ↑ αsyn clearance by microglia.	↓ NeuN cell loss; ↓ NeuN cell loss (hipp); ↑ increased synaptophysin (Hippocampus).	↓ TNF-a and IL-6; ↑ Iba-1 in hippocampus; ↓ Astrogliosis.	↓ latency to turn (Pole test); ↓ Total activity in open field.	[107]
NAC to C-term.	1H7 (IgG1)	91–99 (NAC to C-terminal)	FL h-αsyn	i.p., weekly 6 m 10 mg/kg b.w.	PD: Thy1 αsyn (line 61) mice	↓ αsyn and αsyn aggre. (temporal and striatal neuropil) ↓ axonal αsyn (striatum)	↓ TH loss in striatum; ↑ synapto-physin + MAP2 (neocortex and striatum)	↓ Astrogliosis ↓ Microgliosis	↓ Memory and learning deficits; ↓ error on transversal beam)	[18]
	5C1 (IgG1) (9E4 analog)	C-term. 118–126	GCC-VDPDNEAYE peptide			↓ αsyn and αsyn aggr. (temporal and striatal neuropil) ↓ axonal as αsyn yn (striatum)	↓ TH loss in striatum; ↑ synaptophysin + MAP2 (neocortex and striatum	↓ Astrogliosis ↓ Microgliosis	↓ Memory and learning deficits;	
	5D12 (IgG1) (9E4 analog)	C-term. 118–126	VDPDNEAYE-GCC peptide			↓ αsyn (neocortex)	-	-	-	
	9E4	C-term. 118–126	FL h-αsyn			↓ αsyn and αsyn aggre. (temporal and striatal neuropil) ↓ axonal αsyn (striatum)	↓ TH loss in striatum; ↑ synaptophysin + MAP2 (neocortex and striatum	↓ Astrogliosis ↓ Microgliosis	↓ Memory and learning deficits; ↓ error on transversal beam)	
N/C-term.	Syn303	N-term. 1–5	human phos./nitr. αsyn	i.p., weekly 180 days 30 mg/kg b. w.	PD: Intra-striatal injection of PFF in wt mice	↓ insoluble αsyn aggre. and pS129-αsyn; ↓ Reduced αsyn spread in SNc (30%) and contra- and ipsilateral amygdala (40%).	↓ neuron loss; ↓ PFF neuron entry and PFF transmission; ↓ TH cell loss.	-	↑ latency to hang (Wirehang time)	[108]
	Syn211	C-term. 121–125 (mono./oligo./Fibrils)	h-αsyn positive for DNEAY-peptide			↓ insoluble αsyn aggre. and pS129-αsyn	↓ neuron loss; ↓ PFF neuron entry and PFF transmission.	-	-	
Proto-fibril	mAB47 (IgG1)	Conformational	h-αsyn oligomers (hybridoma)	i.p., weekly 14 wks 10 mg/kg b.w.	PD: Thy-1 H[A30P] mice	↓ αsyn protofibrils in spinal cord	-	-	-	[109]
N/C-term.	AB1	N-term. 16–35	αsyn peptide (16–35aa)	i.p., 14 days 3 m, 1 mg/rat (2x first), then 0.5 mg/mL	PD: Nigral AVV-CBA-αsyn in wt rats	↓ αsyn in SN	↓ DA and NeuN cell loss.	↓ Microgliosis	-	[110]
	AB2	C-term. 93–115	αsyn peptide (93–115aa)			↓ αsyn brain homogenate	-	↓ Microgliosis	-	
Oligo and late aggre.	Syn-01	Conformational (Oligo./aggre.)	αsyn -> hybridomas	i.p., weekly 3 m 30 mg/kg b.w.	PD/DLB: mThy1 αsyn (Line 61) Mice	↓ αsyn (neocortex, hippocampus, striatum, SN); ↓ PK-resistant αsyn (neocortex, hippocampus, striatum); ↓ oligomeric αsyn; ↓ 5G4 aggregated αsyn	↓ NeuN hippocampal loss (CA3); ↑ Synapsin I/Synaptophysin ratio;↓ αsyn/synaptophysin ratio	↓ Astrogliosis; ↓ Microgliosis	↓ beam breaks (total activity)	[111]
	Syn-02					↓ αsyn (striatum);↓ PK-resistant αsyn (hippocampus, striatum); ↓ total αsyn; ↓ oligo. αsyn;↓ 5G4 aggregated αsyn.	-	-	-	
	Syn-04					↓ αsyn (neocortex, hippocampus, Striatum, SN);↓ PK-resistant αsyn (neocortex, hippocampus, striatum); ↓ total αsyn; ↓ oligomeric αsyn; ↓ 5G4 aggregated αsyn	↓ NeuN hippocampal loss (CA3); ↑ Synapsin I/Synaptophysin ratio;↓ αsyn/synaptophysin ratio	↓ Astrogliosis; ↓ Microgliosis	↓ beam breaks (total activity)	
	Syn-F1	Conformational (late aggre.)				↓ αsyn (neocortex, hippocampus, striatum, SN); ↓ PK-resistant αsyn (hippocampus); ↓ oligomeric αsyn.	↓ NeuN hippocampal loss (CA3); ↑ Synapsin I/Synaptophysin ratio;↓ αsyn/synaptophysin ratio	-	↓ beam breaks (total activity)	
	Syn-F2					↓ αsyn (neocortex, striatum, SN); ↓ PK-resistant αsyn (hippocampus); ↓ oligomeric αsyn.	↓ αsyn/synaptophysin ratio	-	-	
Aggre.	1H7	C-term. 91–99	FL h-αsyn	i.p., weekly 3 m 30 mg/kg b.w.	PD: mThy1 αsyn (61) mice, intra-hippocampal inj. of LV-αsyn	↓ axonal αsyn ↑ coloc. of αsyn and microglia	↑ axonal integrity	-	↓ water maze time to localization	[112]
Oligo	Rec47 (mAB47 as in [109])	Conformational, Binding to C-terminal 121–127	h-αsyn oligomers (hybridoma)	i.p., bi-weekly 3 m 20 mg/kg b.w.	MSA: PLP αsyn transgenic mice	↓ soluble and insoluble αsyn (hippocampus) ↓ GCI’s in spinal cord; ↑ pS129 αsyn (SNpc, pontine nuclei and inferior olives) ↑ Co-localization of LCS (autophagy) and p-S129 αsyn.		↓ Microgliosis; ↓ activated MG; ↑ Iba-1 and olig-αsyn co-localization	-	[113]
Aggre.	MEDI1341 (IgG1)	C-term.	Human phage library cloned into IgG1	i.p., weekly 13 wks 20 mg/kg b.w.	PD: mThy1 αsyn (Line 61) mice – intra-hippocampal injection of LV-αsyn	↓ contralateral and ipsilateral αsyn (hippocampus); ↓ contralateral axonal αsyn ↓ αsyn (neocortex) ↓ interstitial fluid αsyn levels ↓ CSF fluid αsyn levels ↓ αsyn positive neurons (neocortex and hippocampus)	-	-	-	[114]
Mono. and Oligo.	nAb isolated from IViG		nAbs isolated from IViG using αsyn column chromatography	s.c., weekly 4 wks Low dosage: 0.8 mg/kg b.w.	PD: A53T tg mice	↓ pS129-αsyn (brainstem) ↓ soluble αsyn (brainstem)		↓ Astrogliosis(Striatum);↑ Microglia and αsyn coloc.	↓ Pole test (time to descend/time to turn).	[115]
				s.c., weekly 4 wks High dosage: 2.4 mg/kg b.w.		↓ pS129-αsyn (brainstem and neocortex) ↓ soluble αsyn (brainstem) ↓ Reduced total soluble and insoluble h-αsyn (brainstem); ↓ fibrillary-oligo. αsyn; ↓ pS129-αsyn/NfL ratio. ↑ Microglia and αsyn co-localization.	↑ PSD95 (brainstem), ↑ synaptophysin (brainstem); ↓ TH cell loss (striatum, brainstem)	↓ astrogliosis(striatum) ↓ microgliosis(striatum); ↓ MCP-1(brainstem).	↓ Pole test (time to descend/time to turn); ↑ Body suspension test (hanging); ↑ Y maze (duration in new arm/new entries).	
Aggre.	Syn9048 (IgG1)	C-term.	hybridoma	i.p., weekly 6 m 30 mg/kg	PD: wt + αsyn PFF (5µg) unilateral inj. in dorsal striatum	↓ αsyn ipsilateral SN; ↓ Contralateral amygdala.	↓ DA cell loss; ↑ DOPAC	-	-	[116]
N-term.	Syn303 ([108])	N-term. 1–5	phos./nitr. h- αsyn			-	↓ TH cell loss (ipsilateral)	-	-	
Aggre.	BIIB054/cinpanemab	N-term: 1–10 (800-fold greater affinity to aggregated αsyn)	Healthy human memory B cells -> clones	i.p., weekly 60, 90 or 100 days 30 mg/kg b.w.	PD: wt C57BL/6JRccHsd mice + αsyn PFF intrastrial inj.	↓ truncated αsyn 6kd (100d)	-	-	↑ Hangwire (latency to fall, 60d)	[117]
					PD: Tg αsyn A53T (M83) + αsyn PFF inoc.	-	-	-	↓ paralysis (7 d) ↓ severe paralysis (5 d); ↓ weight loss (9 d).	
					PD: BAC αsyn A53T + αsyn PFF intrastrial inj.	-	↑ contralateral DAT levels (striatum, 90d)	-	-	
NAC-region	NAC32	53–87	Yeast surface display library of an entire naïve repertoire of human scFV antibodies	Stereotaxis (AAV-NAC32) post 12 wks after αsyn inj. Beh. 4,8 and 12 wks after NAC32 inj.	PD: DAT-Cre rats + AAV-DIO- αsyn in SNpc.	↓ αsyn (25%) (SNpc dorsal).	↓ TH cell loss (SNpc dorsal)	-	↓ Horizontal activity; ↓ Total distance travelled; ↓ Movement number; ↓ Movement time; ↑ Rest time; ↓ Vertical activity	[118]
Aggre. (Oligo/Proto-fibrils)	ABBV-0805/mAB47 for murine experiments	Humanized mAB47, binding to C-term. 121–127	h-αsyn oligo. ->hybridoma, same as prior	i.v., bolus, starting at 2 m old, sampled multiple times. 0.1, 1, 10 mg/kg	wt C57BL/6 mice (pharmacokinetics)	-	0.3% in the brain dose-dependent plasma content	-	-	[119]
				i.p., weekly starting at age 12 m, 10mg/kg	PD: Thy-1-h[A30P] αsyn tg mice	-	-	-	↑ Mean survival from 84 days to 160 days	
				i.p., weekly starting at age 12 m, 20 mg/kg	PD: Thy-1-h[A30P] αsyn tg + 10 µg *gastrocnemius* i.m PFF inj., after mab treatment	-	-	-	↑ Mean survival from 84 days to 95 days	
				Starting 4 wks prior to PFF inj.; weekly mab inj. Prophylactic: 2–4 m, until severe motor deficits, 20 mg/kg	PD: Thy-1-h[A30P] αsyn tg + *gastrocnemius* PFF inj. 1 µg i.m.	↓ soluble and insoluble αsyn (brain); ↓ insoluble pS129-αsyn; ↓ CSF pS129-αsyn; ↓ LB-509 αsyn inclusions (reticular nucleus); ↓ pS129-αsyn inclusion (midbrain).	-	-	-	
				Post 2 wks after PFF inj.; weekly mab inj. Therapeutic: 2–4 m. until severe motor deficits, 20 mg/kg	PD: Thy-1-h[A30P] αsyn tg + *gastrocnemius* PFF inj. 1 µg i.m.	↓ soluble and insoluble αsyn (brain); ↓ insoluble pS129-αsyn ↓ CSF pS129-αsyn; ↓ Dose-dependent soluble and insoluble αsyn (brain); ↓ soluble αsyn at low mab administration (0.25 mg/kg); ↓ insoluble αsyn (5 mg/kg); ↓ LB-509 αsyn inclusions in (reticular nucleus); ↓ pS129-αsyn inclusion (midbrain).	-	-	-	
				weekly 16 wks 20 mg/kg	PD: A53T+/− mice (83) + i.c. (anterior olfactory nucleus) PFF inj.	↓ pS129- αsyn pathology spreading to the contralateral hippocampus (CA1) (58%).	-	-	-	

Abbreviations: PFF: preformed fibrils, αsyn: alpha-synuclein, FL: full-length, h-αsyn: human αsyn, CC: Calpain cleaved, pS129-ayn: phosphorylated αsyn, phos.: phosphorylated, nitr.: nitrated, mab: monoclonal antibodies, mono.: monomeric, oligo.:oligomers/oligomeric, aggre: aggregates/aggregated, m: months, wks: weeks, inj.: injection, i.p.: intraperitoneal, i.m.: intramuscular, i.c.: intracerebral, inoc.: inoculation, b.w.: bodyweight, SNpc: substantia nigra pars compacta, tg: transgenic, wt: wild-type, LV: lentivirus.

**Table 2 biomolecules-12-00168-t002:** Passive immunization candidates currently in clinical trials.

Target (αsyn)	Name	Companies	Antibody/Clone	Binding Site (aa)	Clinical Groups	Current Clinical Phase	Clinical Trial ID
Aggre.	PRX002/(*Prasinezumab*)–PASADENA study	Hoffman-La Roche; Prothena Biosciences Limited.	Humanized IgG1 mab version of murine 9E4	Preferable aggregated αsyn within the C-terminal at aa 118–126 (VDPDNEAYE)	PD patients (H&Y < 2)	Phase II; active; recruitment completed.	NCT03100149
Aggre. (Oligo/proto-fibrils)	ABBV-0805	AbbVie; BioArctic Neuroscience AB	Humanized mAB47 mab	Preferable aggregated αsyn within the C-terminal at aa 121–127 (DNEAYEM)	PD patients (<5 years from diagnosis and H&Y < 3)	Phase I; recruiting.	NCT04127695
Aggre.	MEDI1341	Astra Zeneca; Takeda Pharmaceuticals	Humanized IgG1 mab	Preferable aggregated αsyn within the C-terminal (within the aa 103–129 region)	Healthy individuals (MEDI1341 vs. placebo)	Phase I; recruitment completed.	NCT03272165
Aggre.	BIIB054 (*Cinpanemab*)–SPARK study	Biogen; Neuroimmune	Healthy human memory B cells derived mab	Preferable aggregated αsyn, oxidized at N-terminal aa: 4–10 (FMKGLSK)	PD patients (<3 years from diagnosis and H&Y < 2.5)	Phase II; Terminated	NCT03318523
Aggre.	Lu AF82422–AMULET study	H. Lundbeck A/S; Genmab A/S	Humanized IgG1 mab	Preferable aggregated αsyn within the C-terminal at aa 112–117 (ILEDMP)	MSA-P and MSA-C patients (<5 years from diagnosis, UMSARS ≤ 16, MoCA ≥ 22)	Phase II; recruiting	NCT05104476

Abbreviations: αsyn: alpha-synuclein, mab: monoclonal antibodies, aggre: aggregates/aggregated, aa: amino acids, oligo: oligomers/oligomeric.

## Data Availability

Not applicable.

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
