# Peer review of "Passive Immunization in Alpha-Synuclein Preclinical Animal Models"

_biomolecules, 2022, doi:10.3390/biom12020168_

Round 1

Reviewer 1 Report

The authors described the updated information about alpha-synucleinopathies, which include Parkinson’s disease, dementia with Lewy bodies, pure autonomic failure and multiple system atrophy, and the passive immunization treatments for these diseases to remove the pathological misfolded alpha-synucleins in animal models and clinical trials.The text is written very well with appropriate references. The following minor comments are for making this review paper better.

Minor
1. The authors described that abnormal αsyn is accumulated in the body many years before the diagnosis in body-first synucleinopathies. Please describe the authors’ opinion about the timing of abnormal αsyn detection in the body and the appropriate subjects. 
2. Please describe the authors’ opinion about the balance between the risk and the benefit of the passive immune treatment for the subjects who have no overt disease symptoms. What level is the threshold of the abnormal αsyn in the body for starting the passive immune treatment?
3. The first character of the title in Table 1 should be capitalized, i.e., “passive” should be “Passive” in the title.
4. Please change all “asyn” to “αsyn” in columns of Table 1, e.g. “Target (asyn)” to “Target (αsyn)” in the first column, etc. If it is technically difficult to use “α” in the columns of Table 1, the abbreviation in the table legend should be “asyn” instead of “αsyn” to match the description in the legend to those in the columns. 
5. Please remove the extra space between “of” and “α-synucleinopathies” in subtitle 4 in page 6.

Author Response

Dear Reviewer,

Thank You very much for the careful reading of our manuscript and detailed comments. We have addressed each of your comments in turn, highlighted below.

Reviewer #1 comments:

  1. The authors described that abnormal αsyn is accumulated in the body many years before the diagnosis in body-first synucleinopathies. Please describe the authors’ opinion about the timing of abnormal αsyn detection in the body and the appropriate subjects. 
  • We thank the reviewer for this important question. An important challenge in clinical development of immune-based therapies against αsyn is the absence of a reliable diagnostic tool to observe disease progression and intervention effects. A precise biomarker is not only important for testing the efficacy of immunizations, but also for identifying the most suitable patients for clinical trials. One point could be made for the minus-αsyn patients, as proposed to be the case in Parkin and LRRK2 mutation carriers. By means of new (and more sensitive) detection methodologies e.g. Protein Misfolding Cyclic Amplification (PMCA) and Real-time quaking-induced conversion (RT-QuIC) assays, combined with new imaging-based paradigms, αsyn templating and spreading could eventually be evaluated on a personalized basis. Moreover, a priori templating inhibitory experiments could drive the basis for pre-selection of correct immunotherapy and identify efficacy-positive patients. However, whether the immunotherapy should be administrated prophylactic or therapeutic and at what timepoint immunotherapy would be administrated to offer the best results still needs to be elucidated.  
  • We have corrected this in the revised manuscript (see page 12, marked in yellow). For this purpose, we have added the following reference:

Kalia LV, Lang AE. Parkinson's disease. Lancet. 2015 Aug 29;386(9996):896-912. doi: 10.1016/S0140-6736(14)61393-3. Epub 2015 Apr 19. PMID: 25904081.

2. Please describe the authors’ opinion about the balance between the risk and the benefit of the passive immune treatment for the subjects who have no overt disease symptoms. What level is the threshold of the abnormal αsyn in the body for starting the passive immune treatment?

  • The balance between patients identified with prodromal αsyn-templating phenotype and eligibility of passive anti-αsyn immunization that outweighs potential disadvantages in the absence of symptoms, is an important question. The disadvantages of passive immunotherapy are, besides the high costs, temporary immunity, time-consuming and person to person dose-response uncertainty, that the patients could potentially initiate hypersensitivity reactions and perhaps severe adverse effect. So far, passive immunization clinical trials have shown very few severe adverse side-effects, as compared to active immunization strategies in Alzheimer’s patients leading to severe adverse effects such as meningoencephalitis. However, it seems that the new affitope-technology circumvent these effects. The benefits from passive immunization are that the prodromal patients without overt symptoms could be maintained in that stage before developing severe and often invalidating symptoms. The threshold of αsyn levels or more importantly, the subtype of strains and their templating properties, for immunotherapeutic intervention still needs to be evaluated. The methodologies e.g. PMCA and RT-QuIC in the αsyn research field are still fairly new, thus standardization of these methods between different laboratories are vital. We have elaborated on this point at page 7 highlighted in yellow.

3. The first character of the title in Table 1 should be capitalized, i.e., “passive” should be “Passive” in the title.

  • We thank reviewer 1 for noticing this mistake. We have corrected the title in the revised manuscript.

4. Please change all “asyn” to “αsyn” in columns of Table 1, e.g. “Target (asyn)” to “Target (αsyn)” in the first column, etc. If it is technically difficult to use “α” in the columns of Table 1, the abbreviation in the table legend should be “asyn” instead of “αsyn” to match the description in the legend to those in the columns. 

  • We thank reviewer 1 for noticing this mistake. We have made the corrections in Table 1 of the revised manuscript.

5. Please remove the extra space between “of” and “α-synucleinopathies” in subtitle 4 in page 6.

  • We thank reviewer 1 for noticing this mistake. We have corrected this in the revised manuscript.

Sincerely,

Jonas Folke and Nathalie Van Den Berge                                                                                        

Reviewer 2 Report

There is a very important correction that MUST be made in the first line of the introduction. The authors correctly cite data on alpha-synuclein made by Spillantini but incorrectly cite Polymeropoulos 1997 as including data on other mutations as well as alpha-synuclein duplication, trplications, which were discovered later and should be cited independently. 

Author Response

Dear Reviewer,

Thank You very much for the careful reading of our manuscript and detailed comments. We have addressed your comment below.

1. There is a very important correction that MUST be made in the first line of the introduction. The authors correctly cite data on alpha-synuclein made by Spillantini but incorrectly cite Polymeropoulos 1997 as including data on other mutations as well as alpha-synuclein duplication, trplications, which were discovered later and should be cited independently. 

  • We thank reviewer 2 for noticing this important mistake. We have corrected this in the revised manuscript (see page 1, marked in yellow). For this purpose we have added the following references:

Chartier-Harlin MC, Kachergus J, Roumier C, Mouroux V, Douay X, Lincoln S, Levecque C, Larvor L, Andrieux J, Hulihan M, Waucquier N, Defebvre L, Amouyel P, Farrer M, Destée A (2004) Alpha-synuclein locus duplication as a cause of familial Parkinson's disease. Lancet. 364(9440):1167-9

Singleton AB, Farrer M, Johnson J, Singleton A, Hague S, Kachergus J, Hulihan M, Peuralinna T, Dutra A, Nussbaum R, Lincoln S, Crawley A, Hanson M, Maraganore D, Adler C, Cookson MR, Muenter M, Baptista M, Miller D, Blancato J, Hardy J, Gwinn-Hardy K (2003) alpha-Synuclein locus triplication causes Parkinson's disease. Science. 302(5646):841

Sincerely,

Nathalie Van Den Berge and  Jonas Folke

Reviewer 3 Report

This review provides an update on the hypothesis of different synucleinopathy subtypes and past and ongoing vaccination strategies. The authors focus on some of their recent and exciting work, and further discuss future passive vaccination efforts for preclinical and clinical trials in this context. Attention is given to the role of alpha-synuclein and the transmission hypothesis, no cell autonomous mechanisms are considered (but does not necessarily need to be mentioned for the purpose of this review). I enjoyed reading the manuscript, the text is comprehensive and well-written. In some parts there is still room for minor adjustments (or focus) and some topics might deserve additional discussion.

There are several definitions or descriptions in section 2, (synuclein and pathogenesis) that should be more carefully considered such as “disfavored helical oligomers”, for which it is not clear what these are. Are they kinetically or thermodynamically instable, for which they are short lived? Please also cite the work showing that helical synuclein cannot be isolated and that these findings could not be replicated by other groups.

It is further stated that “aggregation of synuclein is driving force of synucleinopathies”. I agree that synuclein is important contributor, certainly in rare familial cases involving synuclein variants or multiplications, but it I disagree that it is the driver of pathogenesis. There are many other instances where other (genetic) risk factors can influence or potentially trigger disease, which shouldn’t be overlooked.

Next is MSA and the description that GCIs are rarely, if never, observed in people without any clinical features. In addition to MSA-P and MSA-C there are several other subtypes of MSA, that do not fit current clinical classifications but include a less common sacral variant, minimal change MSA, non-motor MSA and incidental MSA of which the latter is a form where there is GCI pathology in the absence of clinical features (doi: 10.1007/s00401-008-0398-7). In the case of minimal change MSA, there is widespread GCI pathology and aside from some focal pathology with no clear corresponding neuro- or oligodendrogliopathy and a general absence of other clinical signs. These cases are indeed not frequent, but they do exist. Further it is written that “MSA patients are characterized by low-density spread of LBs”. Please add the references showing that there is a stereotypical pattern of pathology or staging of Lewy pathology in MSA. Have such pathology studies been performed for MSA or where is this described? In the same paragraph it is not clear what is meant by “synuclein pathology in GCIs differ in permutations”?

Conformational templating does not strictly need to proceed in vivo, this can also be achieved in vitro – meaning that it does not require cell to cell transfer to be defined as such. Also more often used in the context of protein strains.

This is an interesting description, “Besides in the brain, αsyn pathology is also observed in several peripheral organs of LBD and MSA patients many years before diagnosis (Beach et al., 2010; Mendoza-Velás-quezet al., 2019), including in the …” - but these references do not show evidence for synuclein pathology during the prodrome? Are these the correct citations? Have studies looked at synuclein from biopsies with a longitudinal follow up of MSA and DLB patients, before being diagnosed with any of the respective conditions? Within the same lines, there is a similar description later in the manuscript, “The main pre-motor diagnostic marker of body-first PD is the presence of premotor RBD, but also includes (co)existence of orthostatic hypotension, patho-logical 123I-metaiodobenzylguanidine (MIBG) heart scan, and/or peripheral tissue biopsies positive for aggregated αsyn (Horsager et al., 2020)”. What type of peripheral biopsies were done by Horsager et al., and what was the result? Did they fit with disease subtype?

What is the aSyn Origin Site and Connectome model? I don’t believe it is explained in this review? Where was this model first described (no references provided)?

Please include the reference for “Body-first PD patients are characterized by a more rapidly progressing phenotype, with faster motor and non-motor progression and more rapid cognitive decline, compared to brain-first PD patients.”

It is suggested that aSyn pathology might start in the gut in response to an external trigger for the body to brain subtype. It would be interesting to know how the authors think about aSyn pathology being triggered in the brain to body subtype? What could be the trigger here and could there be any shared mechanisms with the body to brain subtype? Any links of the two subtypes with common genetic risk factors?

The subsection focused on ‘Passive immunization strategies in animal models’ would benefit from describing at what point the treatment was started, as it’s mentioned for some but not all studies. For example, it would be interesting to indicate if the treatment was started after there were behavioural deficits and detectable aSyn deposits or earlier as a preventive measure. Even more important if the animal model used in the study received PFFs. Moreover, separating the information in different paragraphs might ease the reading flow.

In the review the authors caution for active immunization strategies. Can the authors comment on the auto immune aspects of PD and how this could impact (active) vaccination strategies? As for passive vaccination, does the existence of natural autoantibodies provide any protection against PD?

There is no Table 1 (or I could not find it). Also provide a table with current active and passive vaccination in clinical trials.

Author Response

Dear Reviewer,

Thank You very much for the careful reading of our manuscript and detailed comments. We have addressed each of your comments in the attached response letter.

Sincerely,

Nathalie Van Den Berge

Jonas Folke

Round 2

Reviewer 1 Report

The authors responded to my comments well and improved the manuscript. I recommend publishing this revised version.